



# Quantifying the carbon footprint of conference travel: the case of NMR meetings

Lucky N. Kapoor[1], Natalia Ruzickova[1], Predrag Živadinović[1], Valentin Leitner[1], Maria Anna Sisak[1], Cecelia Mweka[1], Jeroen Dobbelaere[1], Georgios Katsaros[1], and Paul Schanda[1]

[1]Institute of Science and Technology Austria, Am Campus 1, A-3400 Klosterneuburg, Austria

**Correspondence:** Paul Schanda (paul.schanda@ist.ac.at)

**Abstract.** Conference travel contributes to the climate footprint of academic research. Here, we provide a quantitative estimate of the carbon emissions associated with conference attendance by analyzing travel data from participants of ten international conferences in the field of magnetic resonance, namely EUROMAR, ENC and ICMRBS. We find that attending a EUROMAR conference produces on average approximately 1 ton $CO_2$eq. For the analysed conferences outside Europe the corresponding value is about 2-3 times higher. We compare these conference-related emissions to other activities associated to research, and show that conference travel is a substantial portion of the total climate footprint of a researcher in magnetic resonance. We explore several strategies to reduce these emissions, including the impact of selecting conference venues more strategically and the possibility of decentralized conferences. Through a detailed comparison of train versus air travel — accounting for both direct and infrastructure-related emissions — we demonstrate that train travel offers considerable carbon savings. This data may provide a basis for strategic choices of future conferences in the field and for individuals deciding on their conference attendance.

## 1 Introduction

Reducing the emissions of greenhouse gases is an important challenge to limit global warming (Kikstra et al. (2022)). Despite the increasing awareness, global annual greenhouse gas (GHG) emissions have continued to increase steadily, reaching approximately $59 \pm 6.6$ Gt$CO_2$eq in 2023, which is 62% higher than in 1990 (Intergovernmental Panel on Climate Change (IPCC) and Core Writing Team, H. Lee and J. Romero (eds.) (2023); Crippa et al. (2024)).

Academic research activity also leads to carbon emissions, and the relative importance of facturs such as the production of research consumables (e.g. chemicals), the construction and maintenance of scientific instrumentation and buildings, commuting to the work place, and conference travel have been identified as the activities with the largest footprint (De Paepe et al. (2024); Bull et al. (2022); European Molecular Biology Laboratory (EMBL) (2023)).

The climate crisis is a direct consequence of the quantity of greenhouse gases emitted, and it is, thus, a fundamentally *quantitative* question; naturally, the analysis of causes and possible solutions shall therefore use a quantitative approach. Any meaningful action to mitigate the climate crisis – be it at the level of individuals, organisations or communities, or countries –



is bound to having accurate data: deciding in which fields to make changes requires that one identifies which of our activities
are the largest contributors to our carbon emissions.

According to the International Energy Agency (IEA), the transport sector is responsible for around one fifth of all man-made global GHG emissions.[1] Air travel is estimated to contribute around 2.5% of the global $CO_2$eq emissions, accounting for around 1 billion tons (2021) (Bergero et al. (2023)). The climate impact of aviation extends beyond the directly emitted $CO_2$. When accounting for non-$CO_2$ effects, such as nitrogen oxide emissions, water vapor, and contrail formation at high altitudes,
the sector's overall contribution to global warming is estimated to be around 3.5% to 4% (Lee et al. (2021)).

Given these numbers, one may argue that air travel contributes only little to the overall $CO_2$eq emissions and that removing flights would not solve the climate crisis. However, keeping in mind that ca. 90% of the world's population does not fly (Gössling and Humpe (2020)), the fraction of air travel to the carbon footprint of those who do fly can be substantial. As an example, an out-and-back transatlantic trip emits approximately 4.5 tons of $CO_2$eq. To put this number into perspective, the
International Panel on Climate Change's Special Report (SR15) estimated that the remaining global carbon budget for a 66% chance of limiting warming to 1.5°C is approximately 420 gigatons $CO_2$eq (status: 2017). Even a 1.5 °C global warming limit is projected to result in substantial impacts on natural systems (Masson-Delmotte et al. (2018)). Given the world population of approximately 8 billion, a per-person annual "budget" of ca. 4.5 tons $CO_2$eq can be estimated for 2025, a number that is bound to reach zero in 2050. The current average per-capita $CO_2$eq emissions are of the order of 14 t$CO_2$eq (USA, Australia),
7 t$CO_2$eq (Germany) (source: Our World in Data, for 2023). In light of these numbers, it is clear that e.g. a transatlantic trip to a conference is far from negligible, and it is plausible that flying is a substantial portion of the carbon emissions of academics.

Several studies have analyzed the carbon footprint of academic research in general and of travel in particular, e.g. references (De Paepe et al. (2024); Bull et al. (2022); European Molecular Biology Laboratory (EMBL) (2023)). In this spirit, we decided to analyze the emissions related to Magnetic-Resonance conferences. These conferences are driven by the community, and
as a community, we can consider options to reduce their climate impact. We have collected participant lists of ten major MR meetings over the last 10 years, extracted the presumed travel trajectories of the participants and converted these to carbon emissions. In order to do this conversion properly and realistically, we have also reviewed the conversion factors, including the indirect emissions due to e.g. railway infrastructure. We compare the average per-person emissions of conference attendance to other research-related GHG emissions of a typical magnetic-resonance laboratory.
In search for potential avenues to reduce the carbon footprint of EUROMAR, we find that the choice of the conference location is an important factor for the overall emissions, mirroring previous findings (Orsi (2012); Jäckle (2022)). We explored the possibility to have decentralized (two-site) conferences and find some potential (of the order of one fourth) of reduction.

---

[1]In the following, we will use the term $CO_2$eq, in which gases other than $CO_2$ are considered, too, taking into account their global warming potential.



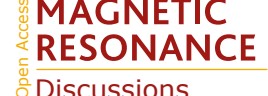

**Figure 1. Distribution of participants at EUROMAR 2024 and ENC/ISMAR 2025 conferences.** A-C Euromar 2024 (Bilbao), D-E ENC/ISMAR 2025 (Asilomar, California). Data were obtained by analysing a list of participants with their affiliations, assuming that the city of the institute of affiliation is the origin of the travel to the conference.





## 2 Magnetic Resonance conference travel in numbers

### 2.1 Distances traveled: past conferences

For our quantitative analysis of conference-travel related emissions we collected data for major MR conferences from 2016 to 2025 (see methods in the Appendix): EUROMAR editions 2016 (Aarhus), 2017 (Warsaw), 2018 (Nantes), 2019 (Berlin, joint with ISMAR), 2022 (Utrecht), 2023 (Glasgow), 2024 (Bilbao); International Conference on Magnetic Resonance in Biological Systems (ICMRBS) editions 2022 (Boston) and 2024 (Seoul), and ENC-ISMAR 2025 (Asilomar, California). These conferences last between 4 and 6 days and hosted between approximately 470 (ICMRBS) and 1100 (joint EUROMAR-ISMAR)
participants; most of the EUROMAR conferences hosted ca. 600–700 participants.

A notable first observation is the geographical distribution of attendees, as illustrated for the case of EUROMAR 2024 in Figure 1. The majority of participants at EUROMAR conferences, often more than 80%, come from Europe. The conference location slightly alters the distribution: we systematically detected additional "local" participants, comprising about 20–30 participants affiliated to the institute of the organisers, as well as more participants from the hosting country. This trend is
particularly pronounced also for the 2024 ICMRBS in Seoul, for example, where 200 of the ca. 580 participants were from the Republic of Korea (Fig. S1); for EUROMAR 2016 (Aarhus), 69 of the 620 were affiliated to a Danish institution, while this number was below 10 for all other EUROMAR editions. Likewise, the ICMRBS 2022 (Boston) showed a strong attendance of participants from the Boston area (75 of ca. 470). This stronger inclusion of the local scientific community is, of course, a desired effect of moving the conference to different places. (We note that for ENC 2025 (Asilomar), this "local" effect is much
less pronounced.)

Figure 2 shows cumulative distribution functions for all conferences (one way distances). It illustrates that for conferences outside Europe, about half of the participants travel several thousand kilometers (roundtrip). Examples of travel-distance distributions are shown in Figure 1C for EUROMAR 2024 (Bilbao) and 1E for ENC/ISMAR 2025 (California). At EUROMAR conferences, the most frequent distance traveled is between 800 and 1200 km, accounting for ca. 40% of the participants.
Another 25% travel distances shorter than 800 km, and the remaining ca. 35% travel longer distances than 2000 km, including long intra-European as well as overseas travel.

### Calculation of $CO_2$eq emissions

To translate this distance information into carbon emissions, assumptions need to be made about the choice of transportation, as well as the respective per-kilometer emissions for trains and planes. (We did not consider car travel.) We assumed, somewhat
arbitrarily, that distances shorter than 800 km are traveled by train, and that for longer distances the participants choose the plane. In doing so, we explicitly used the actual distance of the train trip using the tool *Carbontracer* (https://carbontracer.uni-graz.at/). The assumption that 800 km is a cutoff above which participants tend to prefer plane travel is inspired by analyses of travel data from our institute (reviewing several thousand trips of scientists traveling to/from our institute). Nonetheless, we believe that we might overestimate the number of participants willing to take the train for up to 800 km trips.





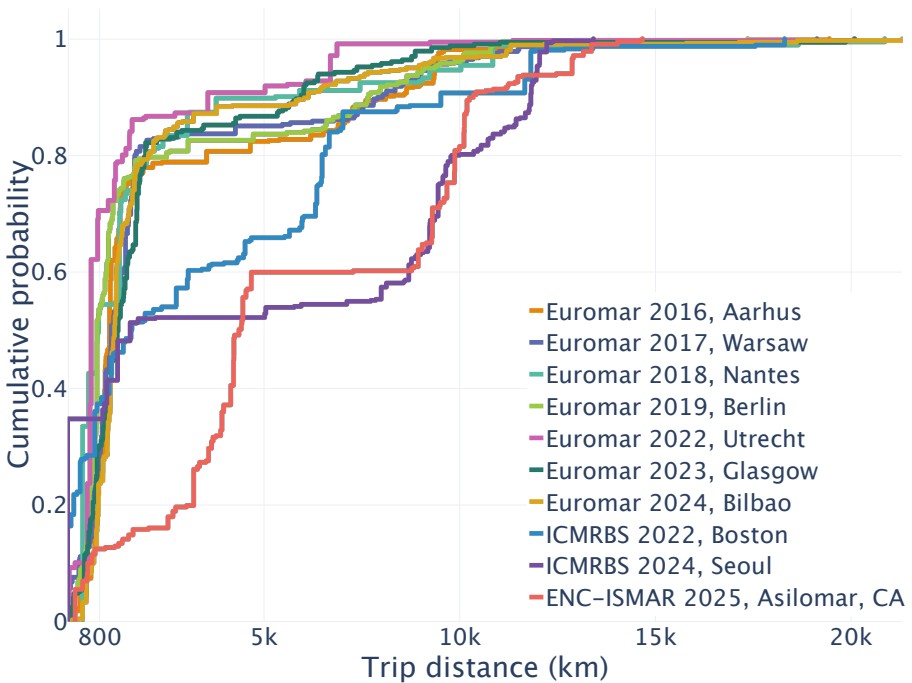

**Figure 2. Distances traveled by conference participants shown as cumulative distribution function, shown as one-way distances. The trip distance is equal to the train distance if shorter than 800 km (i.e. the distance of the rails is explicitly taken into account, using carbontracer (University of Graz (2025)) or to flight distance otherwise.**

For the conversion from kilometers to tons of emitted $CO_2$eq we have performed a literature survey, paying particular attention to using realistic values and including indirect emissions (see Appendix for details). In brief, for air travel, one needs to take into consideration (i) the fact that short distances consume more fuel per kilometer than long-distance flights, (ii) radiative forcing, and (iii) emissions related to infrastructure (airports). Likewise, for train transportation, we sought to obtain a holistic picture that not only includes the emissions related to producing electricity for propelling the trains, but also indirect

effects. In particular, the construction and maintenance of railway lines and buildings are a significant factor for train travel. Details on how we converted distances to emitted carbon are provided in the Appendix. In brief, a value of 25 g of $CO_2$eq per passenger-kilometer in addition to the direct emissions is a realistic estimate for European countries.

    Our calculations have a number of shortcomings, which we shall list here. First, we do not know the mode of transportation chosen by each attendee. Moreover, for flights we assumed direct flights from the airport closest to the participant's affiliation

to the conference location. In reality, many journeys will include connecting flights, which can increase the carbon footprint considerably (of the order of 100 kg (Debbage and Debbage (2019))). Thus, the estimated emissions due to air travel are likely underestimated by ca. 20%. We also ignored possible car travel, which would slightly reduce the emissions (compared to flights) or increase the emissions (compared to train travel). Our estimates of emissions related to train travel are on the





"pessimistic" side, and many websites of railway companies report lower numbers, usually because the indirect emissions are
omitted. We explicitly want to be conservative here and avoid greenwashing of trains (see also below). We also note that the
choice between train and air travel will depend not only on distance but also on the train connections and the availability of
night trains.

Figure 3 shows the total and per-participant travel-related emissions of the analysed conferences. The total travel-related
carbon emissions of EUROMAR conferences were of the order of 700 tons total or about 1 ton $CO_2$ per participant. The per-
person carbon footprint of conferences outside Europe is higher by about a factor of 2-3. This is due to the longer distances
traveled and also the less widespread availability and use of train travel for long-distance travel in e.g. the U.S.

## 2.2 Travel-related emissions dominate the total conference-related emissions

In addition to travel, conference-related carbon emissions are also due to accommodation, catering and the conference site. Data
about emissions of hotels have been collected e.g. by the Cornell Hotel Sustainability Benchmarking (CHSB) Index (Ricaurte
and Jagarajan (2024)), or the Greenhouse gas conversion factors published annually by the UK Department for Environment,
Food & Rural Affairs (DEFRA) or the French ADEME, and are available via web servers such as the hotel footprinting tool
(https://www.hotelfootprints.org/). The emissions scale roughly with the standing of the hotel (due to larger space for higher-
rated hotels). For a 3-star hotel, they are in the range of 10-20 kg per night and person in a European country, which amounts
to several tens of kg for an entire EUROMAR stay.

Meals can be estimated to produce ca. 5.6 kg $CO_2$eq for a meat-based meal, 3.8 kg $CO_2$eq for vegetarian and 2.9 kg $CO_2$eq
for plant based diet (Peter Scarborough (2014)), which amounts to ca. 14-28 kg $CO_2$eq for a 5-day conference. One can
certainly debate whether meals should be counted as conference-specific, as they replace the ones the participants would have
consumed if they were not attending the meeting.

The conference venue requires electricity and possibly natural gas for heating. Although we do not have precise values
for the venues of previous EUROMAR conferences, one can estimate the corresponding carbon footprint to ca. 10 kg per
participant for the entire conference (Educators in VR (2020)).

Overall, we estimate that the GHG emissions of conference attendance other than travel amount to several tens of kg $CO_2$eq.
Thus, transportation to the conference site is the main contributor to the overall footprint of conferences.

## 3 How do these numbers compare to our other research activities?

Our analysis shows that on average attending a typical magnetic-resonance conference produces travel-related carbon emissions
ranging from ca. 0.6 to 2.7 tons $CO_2$ per person, where the variability is primarily due to the conference location; for attending
an overseas conference, the carbon footprint of travel is ca. 4-5 tons per participant. One way of putting these numbers into
context is to compare them to the above-mentioned annual "carbon budget" of 4.5 tons, that is not to be exceeded to limit global
warming to 1.5°C.



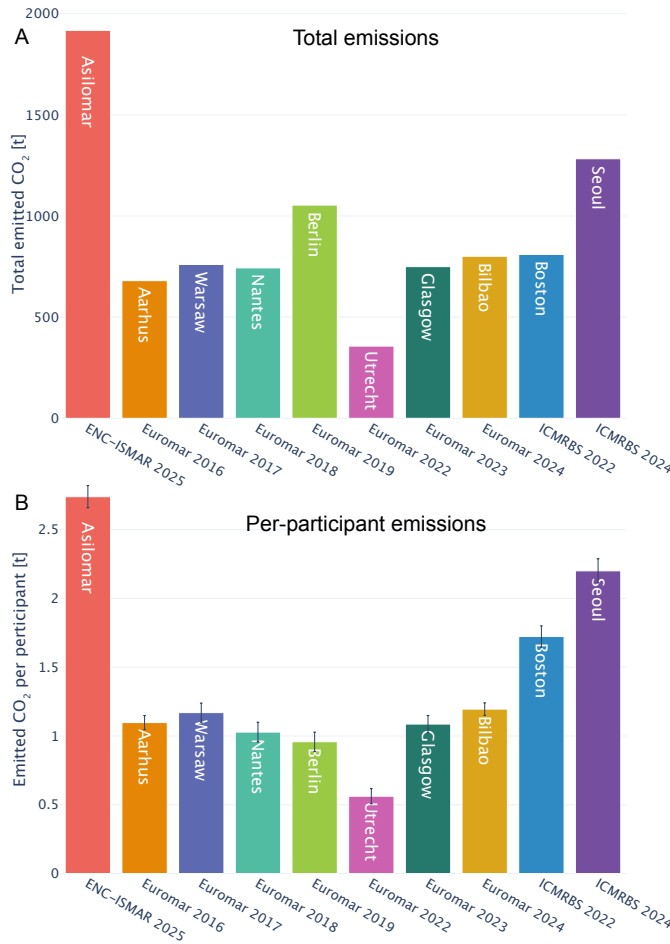

**Figure 3. Travel-related CO₂ emissions from major magnetic-resonance conferences.** (A) Total emissions resulting from the conferences, assuming that (i) participants used air travel if the distance exceeds 800 km and train travel otherwise (for Euromar conferences; for ENC 2025, Californian participants were assumed to travel by car). The number of participants was estimated to be (from left to right): 700, 620, 650, 723, 1100, 635, 690, 670, 470, 583. (B) Per-participant emissions. It is noteworthy that the ICMRBS 2024 in Seoul had a particularly large share of local participants (200 out of 583, see Fig. S1), and the per-person average excluding local participants exceeds 3.1 tons. Similarly, 76 out of the 470 delegates at the 2022 Boston edition were from Boston.

Another interesting way to see these numbers is to compare them to the carbon emissions directly related to our actual research, i.e. everything needed to generate scientific data in the first place, before possibly presenting results at conferences. In light of those, are travel-related emissions possibly negligible anyhow?

     For NMR laboratories, typical activities that generate carbon emissions are: (i) the emissions due to production of NMR machines (supercooled magnets, electronics), (ii) the power consumption to operate these machines and to provide cryogenics, 135    (iii) purchase of computers and running IT infrastructure (e.g. clusters), (iv) construction, maintenance and heating/cooling





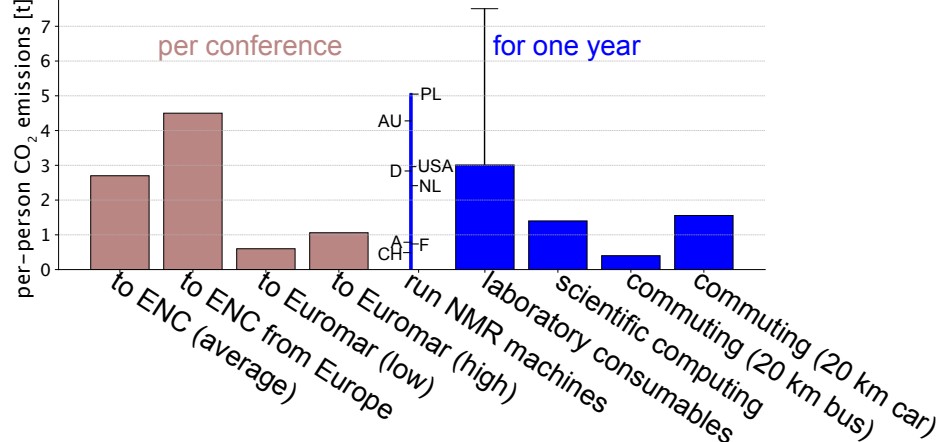

**Figure 4.** Comparison of per-person $CO_2$ emissions from attending conferences compared to those due to activities directly related to research in a magnetic-resonance laboratory, all calculated as per-person emissions, as described in the main text. The emissions for operating NMR machines (three spectrometers used by 20 persons, 600 MHz with solution- and solid-state NMR, 700 MHz solids and 800 MHz with solid- and cryo-probe solution-NMR) were calculated for different countries (Poland, Australia, USA, Germany, Netherlands, Austria, France, Switzerland), and differ because of the different share of fossil fuels for electricity production.

of the buildings we work in, (v) production of samples (e.g. (bio)chemistry laboratory, isotopes, solvents), (vi) commuting to/from work.

For calculating points (i) and (ii), we used data by the R-NMR project online calculator, which considers the power consumption needed for running the console and possibly compressors/pumps, and power related to He and $N_2$ boil-off and liquefaction
(https://csdm.dk/rnmr/consumption.html, version 1.1.5, created by Thomas Vosegaard). As an example, assume a facility with three NMR systems at 600, 700 and 800 MHz (4 K magnets) with solid-state and cryoprobes, located in Austria (the latter will slightly alter the emissions of electricity production). The calculated $CO_2$eq footprint is 8.9 tons for this model calculation. Assuming that this infrastructure is used by 20 group members, the resulting carbon footprint per person is 0.8 tons per year. In other words, participation at a 5-day EUROMAR conference has a larger impact than doing NMR for the entire year. (Figure
4 shows this model calculation also for other countries with more carbon-intense electricity generation.)

Consumables required to produce samples have been identified as the main contributor to carbon emissions in research laboratories (Bull et al. (2022)). In a study involving hundreds of laboratories in France, the emissions related to consumables (i.e. their production, transport and disposal) was estimated to account for ca. 2.7 to 3 tons $CO_2$eq per person (De Paepe et al. (2024)); for a study focusing on chemistry laboratories, a value of 2.3 tons was reported (Estevez-Torres et al. (2024)). There is
a large variability, with values up to ca. 7 tons. We performed our own estimations at our institute, which covers a wide range of fields, using a cost-based conversion metric, and found a value rather at the upper end of this range. Clearly, the exact type of research is a critical determinant of emissions, and these numbers, 2.5-7 tons of $CO_2$eq emissions per researcher, serve as a rough estimate for comparison.





To quantitatively assess the other contributors (computers/IT, buildings, commute), we use data gathered by the ISTA's
sustainability office related to our institute. ISTA is a growing institute at the outskirts of Vienna, and started from zero in
2009; it currently hosts approximately 85 research groups and 1165 employees in total (700 researchers). It spans research in
most fields of natural sciences including experimental groups in physics, chemistry, biology and mathematics, theorists and
machine learning. Its focus is research; teaching is limited to PhD student courses. Electric power in Austria is produced to ca.
83% from renewable resources, which is relevant as the numbers vary for different countries.

At our institute, scientific computing is estimated to have a carbon footprint of ca. 7% of the total $CO_2$eq emissions (power);
another 0.7% is related to the production of the IT hardware; computing/IT together amount to ca. 1.4 tons $CO_2$eq per year per
employee. Note that this number shall be higher in a country with a more fossil-heavy energy mix, such as the U.S. or China
(ca. 60% fossil fuel share) or Germany (ca. 40% fossil) and slightly lower for France (<10% fossil), compared to Austria (ca.
17% fossil).

15% of our carbon footprint is from the electricity we use (2.5 tons) (excluding computing) and 4% from heating our
buildings (0.7 tons). The biggest part of our footprint is from consumables and equipment (41%) (7 tons). Since our institute
is still growing, adding extra lab space comprises 14% of our $CO_2$eq footprint (2.4 tons).

For commuting to work for the entire year (230 days), let us assume a 20 km ride (one way), which results in an annual 2.2
tons if done by car or 0.4 tons by bus.

Figure 4 summarizes these estimates and highlights that the travel-related carbon footprint is by far not negligible. Traveling
to ENC from Europe, for example, emits more $CO_2$ than half a year of making samples, performing NMR and computing
combined. Considering that conducting experiments is the core of our profession and the prerequisite for presenting data at a
conference, it seems evident that traveling may be one of the factors that one may be able to reduce.

## 4 Strategies to reduce conference carbon footprint

In light of these data, what can we do as a community, and as individuals? We believe there are several avenues, which range
from "technical" solutions (e.g. where to host a conference) to more "mindset" approaches.

### 4.1 Comparing train and plane travel

One possibility to reduce the carbon footprint is to choose the mode of transport wisely. In order to establish a solid quan-
titative basis for comparing trains and planes, we have conducted a review of the relevant literature. To explicitly avoid any
greenwashing of trains, we have considered not only the energy required for the transport itself (direct emissions), but also
emissions related to building and maintaining the infrastructure. Moreover, we have explicitly considered the energy mix of
the electricity grid (country-dependent), the actually traveled distance (which is most often longer by train than by plane as
the trajectory is bound to the railway grid), the infrastructure-related emissions for construction, maintenance and operation
(railway network, buildings etc.) and typical passenger occupation of trains (including night train) and planes. Various sets of
assumptions, as well as model calculations based on these assumptions, are shown in the Appendix. Figure 5 shows estimated



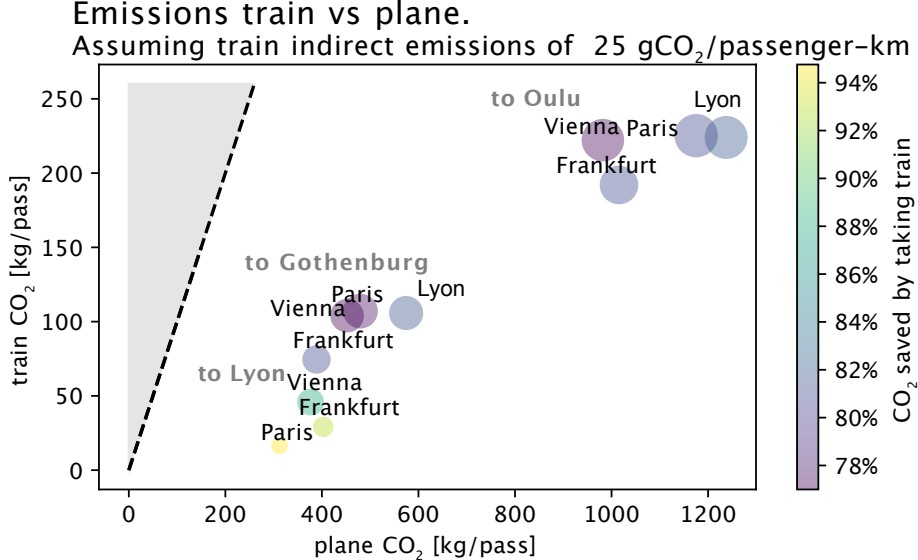

**Figure 5. Assessing the carbon footprint of train and plane travel for a set of chosen destinations (Oulu, Gothenburg, Lyon) and cities of origin.** The x(y) coordinate of each point indicates how much $CO_2$ is emitted on average per passenger by a flight (train). Color of each point shows the fraction of $CO_2$ saved by taking a train instead of a flight and the gray area of the plot corresponds to cases when it is more ecological to take a flight. Finally, the size of the dot is proportional to the ratio of estimated travel time by train and plane between the respective cities ranging from 0.67 for Lyon ⟷ Paris to 8.3 for Oulu ⟷ Vienna

$CO_2$eq emissions of train and plane travel for a set of cities in Europe. As examples, we have chosen journeys from a few European cities (Frankfurt, Vienna, Paris, Lyon) to the upcoming locations of EUROMAR (Oulu, Gothenburg, Lyon). In the case of Oulu, we have also accounted for the ferry transport between Stockholm and Turku. For all these cases, train travel emits much less $CO_2$eq. The reductions range from about 90-95 % (i.e. a 10-20-fold reduction) to about 75 % (4-fold reduction). We
note here that for air travel we have assumed direct flights; stop-overs will add to the emissions (of the order of 100 kg for an additional take-off (Debbage and Debbage (2019))).

These data demonstrate that at the individual traveler's level, the choice of transport is a meaningful way to reduce the carbon footprint. However, for longer distances this is often not a viable option; moreover, the emission reduction also tends to shrink for long distances traveled, in part because train trajectories are longer than the more direct flight trajectories. We also note
that, partly because of political choices, such as the tax exemption of kerosene, train travel tends to be more expensive and train travel may not be possible for this reason.

## 4.2 The choice of the conference location

The significant variability in the per-person emissions of previous conferences (Figure 3) reveals that the location of the conference influences carbon emissions. Naturally, a more central location not only shortens the cumulative distance traveled but





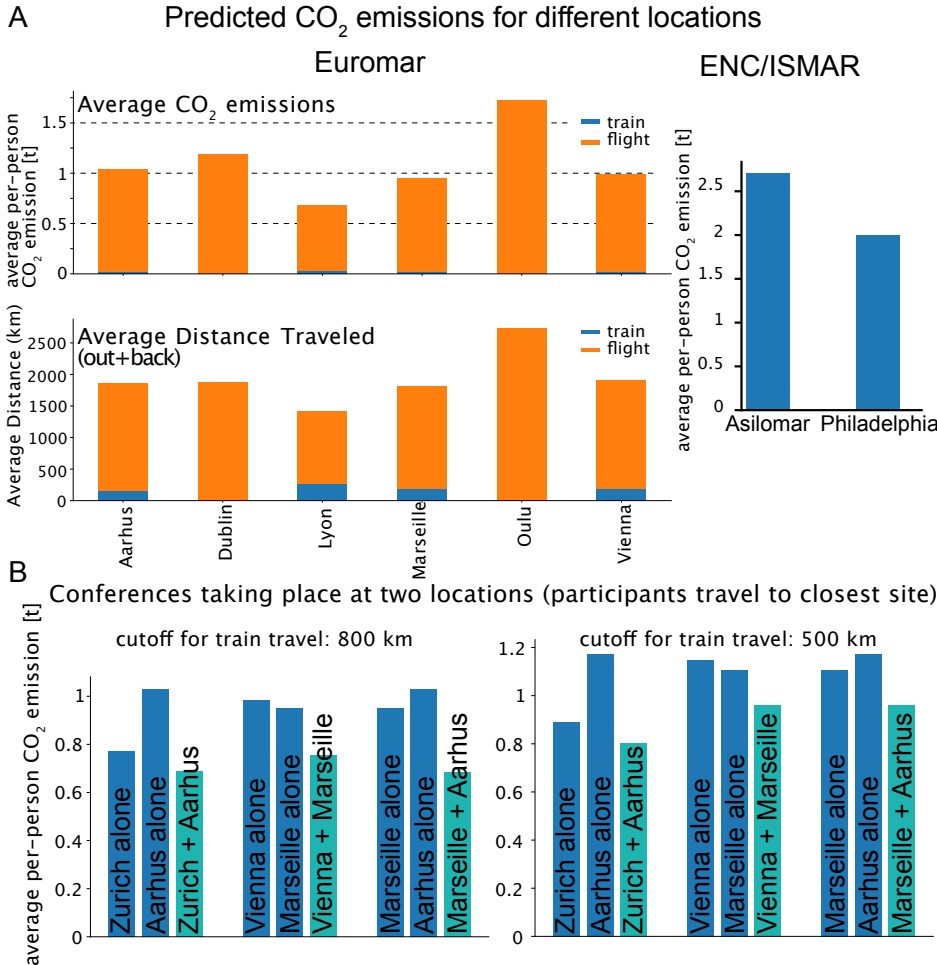

**Figure 6.** Possible strategies to reduce conference-related carbon emissions. (A) Prediction of $CO_2$ emissions for various EUROMAR conference locations. To generate these predictions, we used the list of attendees of the 2024 EUROMAR (removing all but "local" participants from Bilbao). The distances of all attendees to the cities shown here were calculated with *Carbontracer* and converted to $CO_2$eq emissions. The lower panel shows the distances traveled. As in 3, participants were assumed to use train travel if their distance was < 800 km. As a control, the Aarhus conference was predicted based on our assumption, and it matches reasonably well the one calculated from the actual Aarhus list. The panel on the right shows a calculation of ENC, using the participant list of ENC 2025 and performing the calculations if it had taken place at the East coast (Philadelphia). (B) Model calculations for decentralized conferences with two simultaneous locations. The attendee list from Euromar 2024 (as in A) was used to predict the emissions if the conference had taken place in Zurich, Aarhus, Marseille or Vienna (dark blue) and if it had taken place in a joint manner, whereby each participant travels to the conference site closer to their home institution. We assumed that participants would choose train travel up to 800 km (left) or 500 km (right).





also increases the share of train travel. Based on data from previous EUROMAR conferences, we set out to predict average carbon emissions per person for different conference locations (Figure 6A). Choosing a central conference site, such as Frankfurt, Lyon, or Vienna, can reduce the per-attendee emissions by a factor in excess of 2, compared to more remote places such as Northern Scandinavia or Israel. Flight distances to central conference locations are shorter and train travel is feasible for a larger number of participants. For a conference the size of EUROMAR, this means a reduction of the order of 500-800 tons of

$CO_2$. Of course, besides the $CO_2$ savings, it should be noted that colleagues working in places further away from the "center of mass" of the distribution are disadvantaged by systematically choosing more central locations, a factor that shall be considered.

We performed a model calculation for ENC, using the participants from the 2025 edition (which took place in California), and calculated the emission if the conference had taken place on the East coast of the U.S. We predict a ca. 25% saving (400-500 tons overall), which is due to the shorter distance for Europeans and possibly the density of NMR groups on the East

coast.

### 4.3 Multiple parallel virtually connected conference sites

In-person meetings have a quality that online conferences simply cannot provide. Many of us had brilliant ideas (or thought so at that time) while having a drink with a colleague after the poster session. It is not obvious how to generate these opportunities in an online setting. A possible solution is to decentralize conferences by organizing them at several local hubs, which are

connected virtually. Such a conference would then have several parallel sessions – which is common to EUROMAR, ENC and ICMRBS anyhow – and attendees would choose to listen to a talk happening locally or being streamed. A possible modality, presented previously (Orsi (2012)) may look as follows. Initially, the organizers determine the event dates, select a primary location, and identify several optional secondary sites. Next, they publicly announce the conference and main location, and begin accepting participant applications. Then, taking into account the number and geographic distribution of applicants,

participants are allocated to a final selection of venues in a way that minimizes total carbon emissions and maintains suitable attendance levels at each site. Finally, sessions proceed independently at each location, except during key plenary events, which are shared across all venues via videoconferencing. A case study of an international conference predicted this approach to cut emissions by one third with three conference sites (one in Japan, Europe and the USA) (Orsi (2012)).

We have performed a model calculation for a split conference to evaluate the potential reduction in carbon footprint. Figure

6B shows several model calculations for three possible parallel locations. Savings of the order of 25% appear realistic, assuming that participants choose the closer of the two hubs. This number is similar to the one estimated in the study cited in the previous paragraph (Orsi (2012)).

### 4.4 Online-only conference

Online-only conferences also have a carbon footprint, although much lower. The factors that need to be considered include

the participants' devices, the internet infrastructure and the data centers. For example, Zoom/Google Meet/Teams video conferencing results in $\approx 0.150$ to $0.250$ kg $CO_2$eq per hour per participant, which for a 8-h meeting per day amounts to 1.6 kg $CO_2$e per person. This number is two to three orders of magnitude smaller than what we calculated for in-person meetings. A





study of a large astronomy conference, for example, concluded that the online-only version reduced the carbon footprint by a factor of 3000. Clearly, online-only meetings would result in a dramatic reduction in carbon emissions of MR meetings, but
also result in a very different experience.

### 4.5   Less frequent conferences, possibly joint meetings

An obvious way to reduce carbon emissions is to have fewer conferences and/or limit the number of participants per conference, or more precisely, the total distance travelled by participants for all the conferences they attend. Without compromising quality too much, one may achieve this by having meetings back to back at the same location. For example, a EUROMAR may be
preceded by a more specific conference on small molecule NMR, for instance. This concept, in principle, already exists in the form of satellite meetings that often take place before/after EUROMAR meetings. Similarly, we believe that attending a conference for its full duration—rather than leaving early to travel to another—is a meaningful way to improve the benefit-to-footprint ratio.

### 4.6   Embracing more local meetings with fewer long-distance invitees

We believe that it is often seen a mark of success of a meeting to have as wide as possible a geographical distribution of attendees. In light of the need to reduce the carbon footprint, this view shall be reconsidered to adopt a climate-conscious mindset that conferences are best attended mostly by local scientists (local meaning within a country or a continent, which is not that local really), mixed with a small number of international scientists to foster and maintain cross-continental exchange.

### Considering the benefit of conference attendance from a career perspective

On a personal level, choosing to attend fewer conferences is one of the most direct ways to reduce one's carbon footprint. Such a choice, of course, comes with a careful evaluation of the benefits of attending a conference. Legitimate science-based reasons for attending conferences include staying up-to-date with the latest developments, networking with researchers and vendors, initiating collaborations, and increasing professional visibility. These goals appear especially important for early-career researchers seeking academic positions. For example, in a survey of doctoral students and postdocs in Germany, respondents
indicated that conference travel had contributed to collaborative projects and publications; notably, 8% of postdocs reported receiving a job offer as a result (Hauss (2021)). Notwithstanding the potential benefits, studies show that beyond a certain threshold, increased travel does not correlate with higher academic performance (Wynes et al. (2019)). For example, research examining air travel emissions alongside scholarly metrics such as the h-index found no significant relationship between the volume of air travel and research output. This implies that while some conference participation can be professionally beneficial,
excessive travel does not necessarily translate into greater academic success.



## 5   Conclusions

In light of the climate crisis, our society will need to take action quickly and decide how to restructure many aspects of our lives. Climate research, as well as the increased number of extreme weather events, shows that change is not only required but inevitable. This topic is sensitive and we all have opinions on the importance of conferences and their modalities, as well as
the usefulness and necessity of measures to reduce the carbon footprint. Approaching this topic in a spirit of respect and open-mindedness is certainly crucial. We hope to have provided here useful resources for the community and for each individual to take informed decisions and initiate a discussion on changes that the MR community may implement - while keeping an eye on the importance of meeting each other to exchange and progress the field together.

*Code and data availability.*   Python code to perform the calculations, integrating the carbontracer API, are available from the authors upon
request, and will be deposited on IST Austria's Research Explorer repository. Anonymized lists of affiliations are available from the authors upon request.





## 6 Appendix

**Conference travel data used in this analysis**

The analysis of past conferences was based on lists of participants which we obtained either (1) directly from organisers as a
compiled anonymised spreadsheet of affiliations, or (2) as a list available to participants via a conference app, or (3) as abstract
books available from the conference's website. For the latter case, we combined an automatic text extraction tool written in-
house in python programming language using a Google API, with a manual curation. We then performed (largely manual)
internet searches to associate the participants' affiliations to the city in which the participant's institute is located. We assumed
that the participant traveled from this city to the conference site. A python script that calls the *Carbontracer* API was written to
convert start- and end-point of the trip to distances (explicitly taking into account the actual rail or flight distance), as well as the
emissions related to the direct transport. We have added indirect emissions due to infrastructure construction and maintenance,
as shown in the following sections.

**The carbon footprint of train travel**

We estimate the carbon footprint of train travel by adding up the *direct emissions* $D^{(t)}$ of the train journey and the *indirect
emissions* $I^{(t)}$ associated with construction and maintenance of the railway infrastructure and emissions due to heating and
cooling of the station buildings. The index in the exponent of individual variables indicates the corresponding mode of transport,
here $^{(t)}$ stands for train, later $^{(p)}$ indicates plane and $^{(f)}$ represents ferry.

The main contribution of our work is the estimation of $I^{(t)}$. To do that, we use the findings of Landgraf *et al.* (Landgraf
and Horvath (2021)) who estimate the emissions associated with the construction and maintenance of the Austrian railway
network to be $CM^{(t)} = 234730$ tons of $CO_2$eq. We further use information from the OeBB report (Österreichische Bundes-
bahnen (ÖBB) (2022)), which states that the yearly emissions associated with heating and cooling of the station buildings are
around $B^{(t)} = 49500$ tons of $CO_2$eq in 2021 - a number comparable to the total direct emissions of passenger traffic that year
Österreichische Bundesbahnen (ÖBB) (2022).

Further, according to OeBB (Österreichische Bundesbahnen (ÖBB) (2023)), the number of passenger kilometers (pkm)
traveled in 2022 was $T = 11.4$ billion pkm. We take the estimate from 2022 to avoid the effects of decreased travel due to the
Covid-19 pandemic.

Finally, we compute the mean indirect emissions per passenger kilometer:

$$\gamma = \frac{CM^{(t)} + B^{(t)}}{T} = 24.9 \text{ g CO}_2\text{eq/pkm} \tag{1}$$

The above estimate of $\gamma$ is consistent with previously reported estimates of the carbon cost of railway infrastructure (Fig.
5.4, (Tuchschmid et al. (2011))) for several European countries of the order of $8 - 20$ g$CO_2$eq/pkm. Our estimate is on the
higher end and thus conservative when assessing the ecological benefits of train travel.

To generalize our calculations for travel outside Austria, we assume that the construction and maintenance of a kilometer
of rails has a similar carbon footprint in Western and Central European countries. The reasoning is as follows: First, the





rail materials used are the same, and their source is often the same across European countries, thus associated with similar
emissions. Second, (Landgraf and Horvath (2021)) concludes that a substantial amount of the carbon footprint of maintenance
comes from the use of diesel engine vehicles, the footprint of which is also region-independent. We also assume that the amount
of $CO_2$eq emitted per pkm for operating station buildings is, on average, similar across European countries.

We calculate the total emission $E_{AB}^{(t)}$ for a train travel of one passenger from city A to city B as the sum of direct $D_{AB}^{(t)}$ and
indirect $I_{AB}^{(t)}$ emission. $l_{AB}^{(t)}$ is the distance covered by the train between cities $A$ and $B$.

$$E_{AB}^{(t)} = D_{AB}^{(t)} + I_{AB}^{(t)} = D_{AB}^{(t)} + \gamma l_{AB}^{(t)} \tag{2}$$

**The carbon footprint of air travel**

Like train travel, flight emissions consist of direct and indirect emissions. Indirect emissions comprise airport infrastructure
as well as the radiative forcing index ($RFI$), the key emission factor in air travel. The $RFI$ indicates how much more potent
greenhouse gases emitted at a certain altitude are compared to emissions on the ground. Depending on the altitude of plane
travel, the $RFI$ ranges from 1.3 to 3 (Lee et al. (2021)). The indirect emissions for airplanes associated with the construction
and maintenance of airport buildings are estimated to be around $B^{(p)} = 15\%$ of the aviation footprint (Greer et al. (2020);
Sahinkaya and Babuna (2021)). The great circle distance – the shortest distance connecting points A and B on a sphere – is
usually used to calculate the distance of flights. It does not represent real-world flights as it does not account for start- and
landing phases, indirect flight routes, delays, or waiting time in the air. Hence, the great circle distance has to be adjusted by an
uplift factor of 8% for compensation (Department of Business Energy  Industrial Strategy (2017)). The formula for estimating
flight emissions per passenger between cities A and B is:

$$E_{AB}^{(p)} = D_{AB}^{(p)}(RFI + B^{(p)}) \tag{3}$$

where $D_{AB}^{(p)}$ is the direct flight emission per passenger calculated from the estimated flight distance corrected for the uplift
factor.

**The carbon footprint of ferry travel**

Similar to travel by land and air, the emissions of ferry travel comprise direct and indirect emissions. The direct emissions range
from 40 g to 300 g of $CO_2$eq per pkm (Joergen Larsson and Anneli Kamb (2022)). 40 g of $CO_2$eq per pkm seems to be the
most reasonable number since it allocates emissions between freight and passengers in a weight-dependent manner (Joergen
Larsson and Anneli Kamb (2022)). Nevertheless, we used $D_{pkm}^{(f)} = 0.2$ kg of $CO_2$eq per pkm in our exemplary calculations to
stay conservative for our claim that any other transport is more favorable in $CO_2$ emissions than flying. We assumed that ports
are the main contributors to indirect ferry emissions. The greenhouse gas (GHG) emissions at the port of Stockholm were 523
t $CO_2$eq in 2024 (Ports of Stockholm (2024)) while being a hub for 7.2 million passengers. This leads to indirect emissions
of 0.07 kg $CO_2$eq per passenger. Assuming that ports emit the same across Europe and that passengers always use a departure
and an arrival port, it results in $I^{(f)} = 0.14$ kg $CO_2$eq per passenger of indirect emissions per ferry journey. The equation for
calculating ferry emissions is the following:





$$E_{AB}^{(f)} = D_{pkm}^{(f)} l_{AB}^{(f)} + I^{(f)} \tag{4}$$

where $l_{AB}^{(f)}$ is the distance covered by the ferry between cities $A$ and $B$.

**Calculating GHG emissions of different ways of traveling**

As a basis to estimate the total GHG emissions of the different ways of traveling, we used web tools that calculate direct
emissions $D$ caused by a specific trip. To get a range of emission estimates, we evaluate the carbon footprint using multiple
sources of information and multiple alternative settings. We take or calculate $D$ directly from two available online tools:
*Ecopassenger* (International Union of Railways (UIC) (2025)) and *Carbontracer* (University of Graz (2025)).

### 0.1    Ecopassenger

*Ecopassenger* considers GHG emissions that are directly caused by operating the vehicles and the final energy consumption.
For railway travel, the route length is determined by the polygon defined by all train stops on the way to the destination.
The route length between stops is based on the line of sight extended by $20\%$ to $30\%$, depending on the case. *Ecopassenger*
estimates GHG emissions by considering the average national electricity mix of the countries traveled. It allows us to select
the "National production electricity mix" or "Railways mix" and evaluate both scenarios. To estimate the total, infrastructure
including GHG emissions, the resulting direct emissions $D_{(AB)}^{(t)}$ need to be extended as displayed in equation (2).
Flight route lengths are calculated based on the air-line distance, which is corrected for e.g. wait loops by adding 50 km.
Since planes travel longer at higher altitudes when covering longer distances, flights are corrected with $RFI$ factors between
1.26 for distances up to 500 km and 2.5 for distances above 1000 km, but *Ecopassenger* also shows us the value without $RFI$.

    The *Ecopassenger* platform returns the plane travel emissions $P_{AB}^{(p)}$ per passenger for the trip between cities A and B. To
obtain the estimate of total emissions $E_{AB}^{(p)}$, we need to add the infrastructure factor $B^{(p)} = 15\%$. Since the *Ecopassenger*
emissions estimate already includes $RFI$, the resulting equations for $E_{AB}^{(p)}$ is as follows:

$$E_{AB}^{(p)} = P_{AB}^{(p)} + \frac{P_{AB}^{(p)}}{RFI} B^{(p)} \tag{5}$$

where based on (3) the direct flight emissions per passenger are $D_{AB}^{(p)} = \frac{P_{AB}^{(p)}}{RFI}$ and the indirect emissions $I_{AB}^{(p)} = D_{AB}^{(p)} \left( RFI - 1 + B^{(p)} \right)$.

### 0.2    Carbontracer

*Carbontracer* is based on Life Cycle Analysis (LCA) GHG emissions per person per km of the different vehicles. For all types
of travel, it includes not only direct emissions, but also the construction, maintenance, and disposal of the respective vehicle.
Due to the use of various Routing Maps, the actual train routes are calculated. *Carbontracer* considers the national electricity
mixes for train travel and displays GHG emissions per person when traveling seated, in a couchette or in a sleeping car. To
estimate the total GHG emissions, the output given by *Carbontracer* needs to be plugged into the equation (2).





Flight routes are calculated using another set of Routing Maps and compensated by application of the already mentioned
uplift factor of $8\%$. *Carbontracer* independently of flight distance always applies an $RFI$ of 2.

To include airport infrastructure emissions, the flight emissions according to the *Carbontracer* platform $P_{AB}^{(p)}$ need to be corrected as follows:

$$E_{AB}^{(p)} = P_{AB}^{(p)} + \frac{P_{AB}^{(p)}}{2} B^{(p)} \tag{6}$$

The equation takes the same form as (5) with $RFI$ fixed at 2 for *Carbontracer*, thus the 2 in the denominator. Then, analogously
to the *Ecopassanger* section above, the direct flight emissions per passenger are $D_{AB}^{(p)} = \frac{P_{AB}^{(p)}}{2}$ and the indirect emissions $I_{AB}^{(p)} = D_{AB}^{(p)} \left( 2 - 1 + B^{(p)} \right)$.

### 0.3   Results for travel from Vienna to major European cities

Fig. S2 shows the comparison of GHG emissions for plane vs. train travel between Vienna and a number of major European cities that are common destinations of business travel of ISTA employees. The color of each point signifies the fraction of $CO_2$
saved by taking a train. We tested scenarios based on multiple assumptions as follows:

- the direct emissions estimates are obtained from two distinct platforms *Carbontracer* and *Ecopassenger*

- $RFI$ value ranging between 1.3 and 2.7

- electricity mix used by the railways: "national" or "railway" mix

- day train or night train

Plots in S2A show 3 of the most adverse scenarios for trains, where taking a train saves the least amount of $CO_2$eq. It includes cases with low $RFI$ and traveling by night train, which reduces the train's capacity and thus increases the emissions per passenger. Even in those scenarios, a train saves at least $40\%$ $CO_2$eq compared to air travel. S2B shows the representative case scenario, where an intermediate fraction of $CO_2$eq is saved by taking a train instead of a plane. This is the scenario where the $RFI$ is altitude-dependent. Typically, a journey by train saves around $75\%$ of $CO_2$eq. S2C shows the most favorable
scenario for trains when $RFI$ is assumed to be high and railways are assumed to run on a greener electricity mix than the national electricity mix in the respective countries. In such a case, up to $95\%$ of GHG emissions can be saved by avoiding a flight.

Finally, in S2D we show the distribution of the effective factor $\xi$ that is defined as the ratio of total GHG emissions of a train journey and the direct emissions associated with moving the train itself.

$$\xi_{AB} = \frac{D_{AB}^{(t)} + I_{AB}^{(t)}}{D_{AB}^{(t)}} \tag{7}$$

The average value of $\overline{\xi} \approx 2.5$ tells us, as a rule of thumb, by how much one should multiply the GHG emissions estimate by platforms such as *Ecopassenger* to get a realistic $CO_2$ estimate including the costs of railway infrastructure.



$\xi$ ranges from just above one for night trains and journeys to countries with the least clean electricity mix where the indirect
emissions are not larger than the direct emissions; from up to 7 for Austria, where trains are powered exclusively by hydro
power, and therefore the indirect emissions comprise the majority of the total emissions. The estimate for Austria based on
data by Landgraf and colleagues (Landgraf and Horvath (2021)) lands at 6.9.

*Author contributions.* L.N.K, N.R., P.Z., V.L., M.A.S., C.M. and P.S. gathered the data, wrote code for analysis. N.R., V.L., M.A.S., C.M.
developed the approach to calculate indirect emissions. All authors reviewed literature. L.N.K, N.R., P.Z., V.L. and P.S. prepared figures. P.S.
wrote the manuscript with input from all co-authors

*Competing interests.* Some authors are members of the editorial board of journal MR.

*Acknowledgements.* First and foremost, we are grateful to the conference organisers who have provided data, either in the form of tables
or by pointing us to abstract books. This project emerged from an interactive course about energy and climate, held at IST Austria by J.
D., G. K. and P. S. We are grateful to ISTA's Graduate school for enabling this interdisciplinary course and to all participating students. We
thank the following persons for discussions and/or comments about the manuscript: Helene Van Melckebeke, Mei Hong, Jeff Hoch, Gottfried
Otting, Matthias Ernst. For the preparation of the manuscript, AI tools have been used, namely for finding relevant literature (ChatGPT) and
for correcting the text (Writefull, within Overleaf LaTeX).



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
