# Peer review of "Quantifying the carbon footprint of conference travel: the case of NMR meetings"

_Magnetic Resonance, 2025_

## Author Response (AR1)

**RESPONSE TO REVIEWERS**

We would like to thank the reviewers for supporting our efforts to make sustainability part of our thinking. Their comments were really helpful to improve the quality of the manuscript. In preparing this revision we have sought to incorporate their suggestions as far as possible. We believe our manuscript to be substantially improved by these changes and aligned with previous publications. Our novel analysis trying to incorporate all environmental aspects of air and train travel now clearly shows to impact compared to the research we do and provides solutions to reduce the impact of scientific meetings.

Below is our detailed point-by-point response, along with the reviewers' original comments in greens.

**R1 (anonymous)**

I did wonder why 800km was used as the threshold between surface and air travel modes. It is noted that this distance is arbitrary and that it is too high. Klöwer et al (2020) used 400km which is much more realistic. Few will drive or train further than 400km. This figure should be used for consistency with previous studies.

Reply: Thank you for pointing this out. In the version of the manuscript you have reviewed we have used 800 km. We agree that it is high. We have done our calculations again (Figure 3) using a 400 km one-way-distance cutoff. We have additionally kept the 800 km cutoff. Iit shows the savings that one may expect if participants were willing to take train travel up to 800 km. It is interesting to note, for example, that for the conference held in Utrecht there is a sharp drop in emissions with this 800 km scenario. We have cited the study by Klöwer et al (2020), and also added a statement from our institute-internal travel evaluation.

As an aside, the reason for the 800 km cutoff in our original draft was an error in communicating the results from our institute-internal travel evaluation: while our analysis showed that 400 km is a realistic cutoff, we erroneously used the out-and-back distance (800 km) when doing our calculations. We have now corrected this cutoff. The 400 km one-way comes out independently from the Klöwer study and our internal travel analysis.

Some consideration is given to questions of equity particularly regarding career stage (eg early career status). What is missing – and I do think that this is a missed opportunity given the conference discussions that are looming – is acknowledgement of previous research that has addressed alternative conference models and the need for multiple actors to collectively commit to transitioning to a low carbon academic conference convention. Klöwer's (2020) modelling of different conference models including the three hub model that is developed in this paper feels like a glaring omission. The extent to which de-centralised

conferences can reduce emissions should be noted, as this should be a critical focus of discussion and debate at your looming conference.

Klöwer et al's (2020) paper published in Nature in July 2020 really should be integrated into the current manuscript prior to publication as it has so much to offer in terms of conference emissions measurement, emissions modelling under different conference scenarios (and extent of emissions reductions under each), and discussion of pathways forward. Furthermore, in terms of questions of equity there is much qualitative research that informs institutional challenges (eg geographically distant institutions located in the global academic periphery) and barriers to change which again should usefully inform this manuscript and make it more relevant to an international/global readership. See references included at the end of this review as examples. In my view these insights and perspectives should be integrated into the paper prior to publication.

- Klöwer, M., Hopkins, D., Allen, M. & Higham, J.E.S (2020). Decarbonising conference travel after COVID-19. *Nature* 583: 356-360 (16 July 2020) https://www.nature.com/articles/d41586-020-02057-2
- Supplementary Information: https://media.nature.com/original/magazine-assets/d41586-020-02057-2/18168996
- Higham, J.E.S., Hopkins, D. & Orchiston, C. (2019). The work-sociology of academic aeromobility at remote institutions: Networks, co-presence and proximity. *Mobilities* 14(5) October 2019 https://doi.org/10.1080/17450101.2019.1589727
- Hopkins, D., Higham, J.E.S., Orchiston, C. & Duncan, T. (2019). The practice of academic mobilities: bodies, networks and institutional rhythms. *The Geographical Journal* https://doi.10.1111/geoj.12301
- Cohen, S.A., Hanna, P., Higham, J.E.S., Hopkins, D. & Orchiston, C. (2019). Gender discourses in academic mobility. *Gender, Work & Organization* DOI: 10.1111/gwao.12413

**Reply:**

We thank the reviewer for these insightful references, especially *Klöwer et al's (2020)*, which are indeed relevant to our work. We added the suggested references where appropriate, throughout the section *Strategies to reduce conference carbon footprint*. We agree that promoting equal opportunity in the scientific community is an important topic, and we have expanded the section *Online-only conference* to include these important aspects. Our work agrees with *Klöwer et al's (2020)* on many points, including splitting conferences into multiple locations and thus decreasing the number of attendees travelling ultra long distances, which means that a small number of participants contributes disproportionate amount of GHG emissions. We highlight this point in the section *Embracing more local*

meetings with fewer long-distance invitees. Dividing conferences into multiple locations is the focus of our calculations presented in Fig. 6.

**R2 (Jackle):**

In the introduction, the authors point out that air travel accounts for a significant proportion of total emissions because only a very small proportion of the world's population flies at all. Implicitly, the authors ultimately argue for a form of CO2 justice. I could imagine that it would make sense to make this argument explicit, e.g., in the direction of equal per capita shares (Baer et al., 2000; Davidson, 2021).

Reply: we have now added a statement about distributive justice explicitly in the introduction, and we have cited these two references.

As far as I can see, the CO2 emission estimates are correct and easily traceable through the appendix. Since there are a number of possible variables in estimating emissions (e.g., RFI level, different emission factors for electricity production, etc.), it might be useful to present not only the estimated values in Figure 3, but also a range from a minimum to a maximum estimate.

**Reply:**

We agree with the reviewer that given the range of parameters used in our estimations, a range for your estimates should also be indicated. This is the goal of Fig. S2 in the appendix, where the left panels show the "worst" case scenario where taking a train saves the least amount of CO2, while the right panels represent the "best" case scenario, where taking a train would save up to 95% of CO2 for some trips. We considered visualising the variation in the form of errorbars, but the figures became hard to read and for clarity we decided to indicate the range of our predictions this way.

To make this clearer, we highlighted this message in the main text, providing a clear reference to fig S2 as well as highlighting the message of fig S2 by using clear titles.

I very much like the comparison in Figure 4, which shows that a scientist's personal emissions from attending a single conference are similar to those from their normal day-to-day research activities for a whole year. Many researchers probably attend more than one conference a year. If it is known how often scientists in magnetic resonance travel to

conferences on average per year, it would therefore make sense to also indicate the average CO2 emissions caused by the average number of conference visits per year.

We agree with the reviewer that the average CO2 emissions from conferences will change based on the average number of conferences attended. While we do not have data specific to scientists in magnetic resonance, we have analyzed the data for conference/business related travels in our institute. We find that on average Pls travel 2.1 times per year, postdocs travel 1.4 times per year and PhD students travel 0.6 times per year. (The analysis of our institute data is based on the travel bookings from the institute so it underestimates the numbers by not including travels where the host institute handled the bookings / reimbursements.) A more systematic study by Haage (now cited) found that respondents of a survey among German scientists attended 3 conferences per year. We have now mentioned this. While this gives us a multiplication factor, the goal of Fig 4. is to show how the choice of how and where you travel for the conference makes a huge difference on the average CO2 emissions. For example, the CO2 emissions for someone travelling to Euromar as compared traveling to a ENC conference from Europe can be equivalent to more than 4 Euromar conferences attended within Europe. Thus the average CO2 emissions caused by the average number of conference visits will highly depend on the distribution of conferences in a particular year and would need more analysis.

We decided not to change the plot, but rather discuss the fact that many scientists travel to more than one conference per year. This is reflected in the following new/modified statements: "We want to stress here that the data shown in Figure 4 are for a single conference participation, and that the actual annual carbon footprint due to conference travel is likely to be higher for many researchers. A survey of travel behaviour of scientists in Germany has found that respondents of the survey attended on average 3 conferences per year in 2019 (2.2 for PhD students, 4.8 for Pls; \cite{haage2020survey}).

Considering that conducting experiments is the core of our profession and the prerequisite for presenting data at a conference, it seems evident that traveling is a very significant factor of the CO2 emissions of researchers, andm ay be one factor that could be reduced without impacting the scientific output dramatically."

Figure 5 also clearly shows that in some cases (particularly within France, with its high-speed rail network) it is actually faster to travel to a conference venue, thereby saving a significant amount of CO2. However, based on this data, a train journey from Paris to Oulu does not appear to be a very plausible option. To reduce travel-related conference emissions, it might therefore be useful to test (via a survey, see below) how much longer participants would be willing to travel by train rather than fly. In other words, it could make sense to use the factor represented by the size of the dots in the graph as the key variable when planning where to hold a conference.

Reply: We agree that a trip from France to Northern Finland is not necessarily something that many people are willing to do. But we know of participants who did that trip, and sought to calculate how much the savings are.

We agree that a survey among participants/in the community would be useful. This is precisely what the EUROMAR board has launched this summer, and a survey is ongoing. We hope that our analysis will help in this process. We have added the following statements in the Conclusions:

"We hope to have provided useful resources for the community and for each individual to make informed decisions and initiate a discussion on potential changes within the MR community, all while recognizing the importance of meeting to exchange ideas and advance the field together. Studies like the present one will benefit from more targeted data: in particular, it will be useful to directly ask participants/the MR community about their travel habits, preferences, and willingness to reduce the community's carbon footprint."

In addition to the options for reducing emissions mentioned by the authors, hybrid conference models could also be used, in which, in the best case scenario, those who would otherwise have to travel very long distances by plane are connected online, while those who can travel relatively CO2-neutrally by train meet in person. Given the extremely uneven distribution of emissions due to the small number of participants who have to travel extremely long distances, this would lead to a significant reduction in emissions. At the same time, the conference experience for the majority of participants would not differ greatly from a traditional on-site-only conference.

**Reply:**

We agree with the reviewer that reducing the number of attendees from far away is one of the most effective strategies for reducing the carbon footprint of international conferences. We suggest a similar scenario in the section *Embracing more local meetings with fewer long-distance invitees*, where we propose that most attendees of a conference would come from closeby and could travel sustainably, with a few participants from other parts of the world to foster inter-continent exchange. If these participants joined online, as per your suggestion, this would save even more CO2, and we included this case in the section *Embracing more local meetings with fewer long-distance invitees*.

I would encourage the authors to really weigh up the undoubtedly positive aspects of on-site conferences against the emissions they generate, and to compare this with the far lower CO2 emissions of online conferences. According to my own calculations, European political science conferences generate at least 200 times more CO2 emissions than online conferences (Jäckle, 2021). Against this background, it must be asked whether an on-site conference really generates 200 times better output for the advancement of science than an online conference (because that should be the main criterion in the scientific field). The authors do explain this fact (even using the example of an astronomy conference where the savings from online conferences were estimated to be even greater, 3000x), but then they simply conclude that online conferences offer a very different experience from on-site

conferences. This seems a little simplistic to me. Furthermore, online conferences may also offer a lot of positive effects which are often not discussed, (e.g. easier to attend for family care givers, cheaper to attend which increase the inclusion of young scholars and scientists from the global south). Discussing online conferences always only as suboptimal alternatives to in person conferences is thus in my view too short-sided.

**Reply:**

We thank the reviewer for this suggestion and the useful literature reference. Indeed, the difference between online and on-site attendance is worth elaborating on, and we have expanded the section *Online-only conference*' by discussing the equity aspects of online conferences, citing the suggested reference as well as previous work on these aspects.

We assumed, somewhat arbitrarily, that distances shorter than 800 km are traveled by train, and that for longer distances the participants choose the plane." (line 79). A better option might be to use the actual travel time needed as a tool for deciding on a transport option. For example, within France, the high-speed TGV rail network enables fast travel (e.g. from Lille to Toulouse, a distance of about 900 km by car, in 6 hours 40 minutes). In this case, travelling by rail is clearly an alternative to flying. In other countries, however, the same distance by train would take a whole day.

**Reply:**

We thank the reviewer for this comment.

We have modified this part, as outlined also in the reply to reviewer 1. In particular, we have corrected our 800 km cutoff to a 400 km cutoff (one way), in line with the study by Klöwer, and also in line with our institute-internal travel data.

Indeed, travel time is an important factor when people decide how to travel, and could be used as a cutoff criterion instead of distance travelled. However, we decided for distance as a criterion, for the following reasons. First, our information about travel habits came from data of ISTA employees travelling from Vienna to most common conference locations, typically big European cities west of Vienna. According to tools such as Chronotrains (<a href="https://www.chronotrains.com/en/station/2761369-Vienna?maxTime=5">https://www.chronotrains.com/en/station/2761369-Vienna?maxTime=5</a>), the distance one can travel from Vienna within a certain amount of time is very similar for the major cities. Therefore, in our data, it was not possible to distinguish between distance and time when understanding travel habits. The second reason is that convenient travelling is via night trains in which case the time of travelling can be much longer than for the same journey taken in a day. In such cases, the time of travel can be misleading.

We would argue that these estimates are never perfectly precise. The comparison of the 400 km cutoff with the 800 km cutoff (Figure 3) gives the reader an estimate for how these numbers would change with different assumptions.

We added the discussion on using time of travel as a criterion, as well as the possibility of acquiring more data about the travel habits of conference participants via questionnaires in the section *Calculation of CO2eq emissions*.

Furthermore, it probably makes not much sense to include train travel in the estimations for Boston, MA and Pacific Grove, CA, since it is unlikely that anyone travelled by train to these conferences. It would be much more realistic to assume car travel here.

We agree that, given the train network in the US, train travel is certainly less relevant, and we had not included train travel for these conference (possibly this was not stated clearly). We have made this more explicit now: "For conferences outside Europe we did not consider train travel; for participants within 400 km to Asilomar or Boston we assumed transport by car."

"We assumed that the participant traveled from this city to the conference site." (line 278). While this assumption is reasonable and is used as the basis for many similar analyses, a short online questionnaire asking conference participants how and from where they travelled to the venue could be used in future research. The scientific conference organisers could send such a questionnaire to increase response rates.

This is a very good suggestion, and indeed, as mentioned above, the EUROMAR board has launched such an initiative, and in our Conclusions section we now explicitly encourage to perform such surveys.

When calculating the direct emissions of railway travel, the authors base their estimations on Austrian data. However, the Austrian emission factors for the production of electric energy are probably not representative of the rest of Europe (they are much smaller due to the high amount of hydro energy). To get a more realistic estimate of the direct emissions from railway travel, country-specific emission factors for electricity production should be used. Alternatively, in countries where railways are not yet electrified (e.g. Great Britain), the emission factors for diesel locomotives should be used (e.g. Switzerland is almost 100% electrified, whereas in Ireland it is only 2.6%:

https://www.statista.com/statistics/451522/share-of-the-rail-network-which-is-electrified-in-europe/).

**Reply:**

Indeed, the national energy mixes across Europe vary a lot and range from 8 g CO2eq/kWh in Sweden to 594 g CO2eq/kWh in Poland (2023,

https://www.eea.europa.eu/en/analysis/indicators/greenhouse-gas-emission-intensity-of-1). Both tools, Ecopassenger and Carbontracer, used in this work to calculate the direct emissions of railway travel, do consider these differences and calculate the emissions according to the distance travelled in the respective country. We extended those direct emissions by the factor of indirect emissions that comprise the construction and maintenance of railway infrastructure, as well as operating station buildings. To calculate those indirect emissions, we assumed that the construction and maintenance of a kilometer

of rails has a similar carbon footprint in Western and Central European countries. We justify this assumption since rail materials are the same across Europe and often share the same source and therefore can be associated with similar emissions. This assumption does not take into account the different geographies across the continent, since building a train track on flat land is less CO2-intensive than a track that leads through mountains and requires tunnels and bridges. Since we took the more mountainous Austria to calculate these indirect emissions, we are conservative in this regard. We also assumed that the amount of CO2eq emitted per pkm for operating station buildings is, on average, similar across European countries. However, we did not investigate further how dense the networks of station buildings in different countries are or how CO2-intensive their heating and cooling, respectively, are.

We have added the following statement in the Appendix:

"While tools exist for calculating the direct emissions (i.e. the production of energy for moving the trains), including the different carbon intensities of electricity production in different countries (see section "Calculating GHG emissions of different ways of traveling"), we wanted to obtain a reliable estimation of I(t)."

and a statement in the main text:

"The carbon emissions for production of electricity vary significiantly for different countries (from 8 g CO\textsubscript{2eq}/kWh in Sweden to 594 g CO2eq/kWh in Poland (2023, \cite{EEA2025GHGIntensity}). We have considered these differences, see Appendix."

**Additional changes:**

We have updated figures 3 and 4, in order to account for the 400 km (one-way) cutoff of train travel.